# Classifying Circumnutation in Pea Plants via Supervised Machine Learning

**DOI:** 10.3390/plants12040965

**Published:** 2023-02-20

**Authors:** Qiuran Wang, Tommaso Barbariol, Gian Antonio Susto, Bianca Bonato, Silvia Guerra, Umberto Castiello

**Affiliations:** 1Department of General Psychology, University of Padova, 35132 Padova, Italy; 2Department of Information Engineering, University of Padova, 35131 Padova, Italy

**Keywords:** plant movement, circumnutation, machine learning, classification, kinematics

## Abstract

Climbing plants require an external support to grow vertically and enhance light acquisition. Climbers that find a suitable support demonstrate greater performance and fitness than those that remain prostrate. Support search is characterized by oscillatory movements (i.e., circumnutation), in which plants rotate around a central axis during their growth. Numerous studies have elucidated the mechanistic details of circumnutation, but how this phenomenon is controlled during support searching remains unclear. To fill this gap, here we tested whether simulation-based machine learning methods can capture differences in movement patterns nested in actual kinematical data. We compared machine learning classifiers with the aim of generating models that learn to discriminate between circumnutation patterns related to the presence/absence of a support in the environment. Results indicate that there is a difference in the pattern of circumnutation, depending on the presence of a support, that can be learned and classified rather accurately. We also identify distinctive kinematic features at the level of the junction underneath the tendrils that seems to be a superior indicator for discerning the presence/absence of the support by the plant. Overall, machine learning approaches appear to be powerful tools for understanding the movement of plants.

## 1. Introduction

When observing plants, they seem relatively immobile, stuck to the ground in rigid structures. But for careful observers, as Darwin was in the 19th century, it is quite clear that plants do produce movement. Darwin was fascinated by the graceful movements of twining plants revolving in large arcs, winding around a support, and forming a helical tube of tissue. He described this movement as “a continuous self-bowing of the whole shoot, successively directed to all points of the compass” [1] and later named this movement circumnutation [2]. 

Circumnutation is a common phenomenon in plants but is exaggerated in twining stems. By circumnutating, twiners increase the probability of encountering a support to grow vertically and enhance light acquisition. Vines that find a suitable support have greater performance and fitness than those that remain prostrate [3,4]. Therefore, locating a suitable support is a key process in the life history of climbing plants. Numerous studies on climbing plant behavior have elucidated mechanistic details of support searching and attachment [e.g., 3]. This body of research relies chiefly on field observations reporting on morphological or physiological responses [4], as well as on laboratory studies focused on the characterization of kinematical patterning through the use of time-lapse photography [5,6]. Although this body of research provides some quantitative data, the process is admittedly subjective and rather preliminary. In other words, it does not offer a clear explanation of what happens in the pattern of circumnutation when climbers perceive a potential support and decide to orient their movement towards it. 

Machine learning approaches might be an alternative method of addressing this issue and enabling accurate phenotyping. The application of machine learning to questions in plant biology is still in its infancy, yet the applicability of these methods to a broad range of problems is clear. These technologies have recently achieved impressive performance on a variety of predictive tasks, such as species identification [7,8], plant species distribution modeling [9,10], weed detection [11], and mercury damage to herbarium specimens [12]. They are also being applied to questions of comparative genomics [13], gene expression [14], and to conducting high-throughput phenotyping [15,16] for agricultural and ecological research. Machine learning methods, however, have never been used for modeling or predicting the movement of plants. Predicting plant behavior through their movement is important for several reasons. Realistic predictions could aid in the formation of conservation strategies to combat the decline in biodiversity. For example, predicting movement might be important in the context of understanding the spread of infectious diseases through plant species. Many diseases are spread through different means of communication between individuals. Realistic predictions of the movement of infected individuals can suggest interventions that will optimally alleviate the further spread of diseases.

In this connection, here we use machine learning methods to classify plant movement behavior, and to predict movement patterns which will enable us to build stochastic movement generators, useful in scenarios where collecting actual movement data is laborious. Given that predicting plant movement is important when building simulators, we tested whether simulation-based machine learning methods can capture the movement patterns nested in actual kinematical data. We compared several machine learning classifiers to model plants’ movement with the goal of generating models that, on the basis of a binary labeled dataset, learn to discriminate between the presence/absence of a support in the environment so as to formulate precise predictions based on an unlabeled dataset. We found that there is a difference in the pattern of circumnutation that can be learned and classified rather accurately depending on the presence or absence of the support. Furthermore, we identified the most distinctive kinematic features that contribute to the classification tasks and provide additional information for driving future circumnutation studies. Overall, machine learning appears to be a valid tool for studying the movement of plants.

## 2. Results

### 2.1. Classifiers Are Able to Perform Accurate Predictions Depending on the Presence/Absence of the Support

To test whether the pattern of circumnutation differed depending on the presence/absence of a support in the environment, we exposed pea plants to a condition in which a support was not present in the environment (“no support” condition; Figure 1a) and a condition in which a support was present in the environment (“support” condition; Figure 1b). The plants that grew in the presence of the support oriented their movement towards it and prepared for grasping. The plants that grew in the absence of the support continued to circumnutate toward the light source and then fell down. From the 3D reconstruction of movement trajectories for the considered anatomical landmarks (i.e., the tendrils and the point where the tendrils tie, from now on “junction”; Figure 1c), we extracted a set of kinematic features that were used for machine learning classification (see details in Section 4 Material and Methods; also see Appendix A). Three classifiers, namely random forest (RF), logistic regression (LR), and support vector classifier (SVC), were used as a cross-model validation [17]. These approaches have been optimized and validated in a variety of fields [18,19]. The classifiers generated models based on a binary-labeled training set, learned to discriminate between the presence/absence of the support, and formulated precise predictions based on an unlabeled test set. The performance corresponds to the accuracy of classification (i.e., the rate of discriminating plants growing in the presence/absence of the support on the test set correctly). When considering the totality of the circumnutations performed by the plant (i.e., “overall movement classification”), the classifiers were able to distinguish between the “support” and the “no support” conditions with a mean accuracy across all classifiers and all features of 66.80 % (SD 15.39; Table 1). When considering circumnutations singularly (i.e., “circumnutation classification”), the mean accuracy was 68.52% (SD 12.63; Table 2). These results demonstrate that the classifiers were capable of differentiating the pattern of circumnutation depending on the presence/absence of the support rather accurately above the chance level (>50.00%).

### 2.2. Specific Contribution of the Considered Features across Classifiers for the Overall Movement Classification

As shown in Table 1 (also see Appendix A), the SVC performs with a slightly higher average accuracy (mean 67.60%, SD 11.94) compared to the RF (mean 66.30%, SD 17.36) and LR (mean 66.40%, SD 16.37) classifiers. Regarding those features that contributed to the successful classification, the “junction velocity” (mean 77.30%, SD 11.99), the “junction trajectory” (mean 74.30%, SD 14.80), and “all features” (mean 73.20%, SD 12.27) show generally better performance compared with the “tendril aperture” (mean 57.30%, SD 13.17), the “tendril acceleration” (mean 57.2%, SD 12.01), and “movement duration” (mean 56.70%, SD 16.48). With a mean accuracy of 80.50% (SD 13.54) obtained with the LR classifier, “junction trajectory” seems to be the best indicator for distinguishing between the “support” and “no support” conditions. Overall, this suggests that the plants exhibit differences in behavioral patterns depending on the presence/absence of the support.

### 2.3. Specific Contribution of the Considered Features When Considering Single Circumnutations

On the basis of the features derived from a single circumnutation, the classifiers can reliably predict whether the plants are moving in the presence/absence of a potential support (Table 2; also see Appendix A). In comparison to the RF (mean 66.20%, SD 11.60) and the SVC (mean 70.29%, SD 12.98), the LR has a slightly greater average accuracy (mean 69.07%, SD 12.96). As for the contribution of the different features, “all features” (mean 73.08%, SD 11.51), “junction trajectory” (mean 72.75%, SD 12.29), and “tendril velocity” (mean 69.52%, SD 13.63) exhibit better performance compared with “tendril trajectory” (mean 65.17%, SD 12.61), “tendril aperture” (mean 65.03%, SD 11.82), and “tendril acceleration” (mean 64.90%, SD 11.29). With a mean accuracy of 74.87% (SD 12.14) obtained with the LR classifier, “junction trajectory” seems to be the best indicator for distinguishing between the “support” and the “no support” conditions. This is in accordance with the findings for the “overall movement classification.” Again, this demonstrates that the classifiers are able to extract from the kinematics of circumnutation whether the plant is moving in the presence/absence of a potential support.

### 2.4. The Accuracy of the Classification Depends on Organs and Features

When looking more deeply into the contributory role played by the features considered for classification, we found that kinematic features for the tendrils appear to be less relevant with respect to junction-related features for both classification tasks (Table 1 and Table 2). When considering movement duration, this feature appears to be less informative when the overall movement classification is considered. However, this very same feature appears to be a reliable indicator when single circumnutations are considered (68.71%, SD 13.02).

### 2.5. A Full Kinematic Profile Favors Classification

When we combined all the extracted features, we achieved a high level of accuracy across all classifiers (overall movement classification: mean 73.20%, SD 12.27; Table 1; circumnutation classification: mean 73.08%, SD 11.51; Table 2). After the models had been fitted, the importance of kinematic features was determined by applying permutation importance (Figure 2a,b). Different feature importance is detected among classifiers when considering overall movement and single circumnutations separately. For instance, when the overall movement is considered, “junction velocity”, “junction trajectory”, and “junction acceleration” appear to be the most crucial classification features, whereas “tendril acceleration”, “tendril aperture”, “tendril trajectory”, and “movement duration“ appear to be less essential. The negative value (<0.00) for the less important features mentioned above indicates that predictions based on shuffled data typically turn out to be more accurate than real data. “Junction trajectory” and “junction acceleration” appear to be more important than “tendril acceleration” and “tendril aperture” for classification when single circumnutations are considered. Movement duration is an important feature for distinguishing between the presence/absence of the support when it is referred to single circumnutation, but not when “overall movement duration” is considered.

## 3. Discussion

In this study, we propose a general framework to classify pea plants’ circumnutation movement. We applied this framework using various machine learning models ‘fed’ with kinematic data. Our findings show that machine learning techniques have the ability to unveil how kinematic patterning is modulated in key organs when pea plants ‘hunt’ for a support.

Nutation kinematics of different organs has served to lay a foundation of several mechanisms postulated as responsible for the movement in question with tendrils being amongst the most investigated [3,20,21,22]. Tendrils serve climbing plants by providing a parasitic alternative to building independently stable structural supports, allowing the plant to wend its way to sunlight and numerous ecological niches [23]. Accordingly, previous evidence provides a degree of support that some climbing plants can modify their *circumnutation* patterns to a greater or lesser extent depending on the presence/absence of a potential support in the environment [24,25]. Experimental evidence demonstrating that this is the case has been forthcoming from recent studies that used kinematic analysis to characterize the movements of the tendrils of pea plants [6,26,27,28]. Guerra and colleagues [29], for example, demonstrated that pea plants (*Pisum sativum* L.) are able to perceive a stimulus and modulate the kinematics of the tendrils according to the features of a potential support. Therefore, it seems that the tendrils of climbing plants reaching to grasp a stimulus plays a pivotal role as far as support detection is concerned.

The findings of the present study, however, seem to suggest that, rather than the tendrils, the junction underneath them is a superior indicator for discerning the presence/absence of the support. The fact that the kinematics of the junction is a stronger predictor than the kinematics of the tendrils for the presence of the support points to this organ as a navigator guiding the tendrils towards the support. Indeed, if we look carefully at how circumnutation unfolds once the support has been somewhat detected, it is evident that the junction of the tendrils modifies its velocity and timing to launch the tendrils toward the support. In addition, once informed that the ‘take-off’ is approaching, the tendrils open and assume a choreography so as to accommodate the thickness and the shape of the support [29]. All of this corroborates the idea that plant movements are adaptive, flexible, anticipatory, and goal-directed. Put simply, they are somewhat organized and structured, with different organs ‘cross-talking’ to accomplish a crucial endeavor for the plant’s survival. Our study using machine learning techniques illuminates and quantifies this proposal. 

Another novel observation that stems from our study is the classifiers being able to extract a tremendous amount of information from a single circumnutation, which represents the smallest unit of the entire movement. The very fact that the classifiers can make accurate predictions from the emergence of the very first circumnutation reveals that the plants, at the time they initiated to circumnutate, were already well-aware of their surroundings. 

At this stage, a central question that could be asked is whether climbers actually make ‘decisions’ when it comes to support searching and selection. In this respect, our study supports the notion that climbers do not find support/hosts merely by chance. Apart from the evidence of oriented growth towards experimental stakes as discussed above, the methodology used here might be useful to understand climbing plants’ preferences. This was first described by Darwin for tendrils in *B. capreolata* initially seizing but then loosing sticks that were inappropriate [1]. A similar phenomenon is observed when herbaceous twining vines get in contact with a very thick trunk and wind up on themselves instead of attempting to twine around it. In the case of annual vines, Darwin remarked that, even without support diameter constraints, it would be maladaptive to twine around thick—and hence large—trees, as these vines would hardly reach high-light layers by the end of the growing season [1].

Further machine learning research should aim at characterizing how circumnutation changes as far as support characteristics are concerned. Predictions and modeling of the cost-benefit analysis of climbing plant behavior should be helpful to infer the selective pressures that have operated to shape current climber ecological communities. In addition to plant movement, as a direct reflection of plants’ internal state, other physiological markers could be added to obtain a more complete, reliable, and consistent picture of how the environment shapes climbers’ behavior. Such technologies will enable the investigation of unknown aspects of the helical growth performed by the tendrils and their junction on an evolutionary scale, shedding some light on the mechanisms involved in the acquisition and evolution of the climbing habits of terrestrial plants.

## 4. Materials and Methods

### 4.1. Subjects and Materials

A total of 32 snow peas (*Pisum sativum* var. Saccharum cv Carouby de Maussane) were chosen as study plants. For each pot, 6 seedlings were potted at 8 cm from the pot’s border and sowed at a depth of 2.5 cm. Once germinated, one healthy-looking sprout was selected and randomly assigned to the experimental conditions: 19 plants were grown individually in chambers without the presence of a support (“no support” condition; Figure 1a), while 13 plants were grown individually in chambers where a potential support was present (“support” condition; Figure 1b). Sprouts were placed 8 cm from the pot’s border and sowed at a depth of 2.5 cm. The support was a wooden pole (54 cm in height and 1.3 cm in diameter) inserted 7 cm below the soil surface and positioned 12 cm away from the plant’s first unifoliate leaf. 

### 4.2. Growth Setup

Each plant was positioned in a thermo-light-controlled growth chamber (Cultibox SG combi 80 × 80 × 160 cm; Figure 1). The temperature was set at 26 °C by means of an extractor fan equipped with a thermo-regulator (TT125 vents; 125 mm-diameter; max 280 mc/h) and an input-ventilation fan (Blauberg Tubo 100–102 m^3^/h). The two-fan combination allowed for a steady air flow rate with a mean air residence time of 60 s. The fan was carefully placed so that air circulation did not affect the plants’ movements. 

Cylindrical pots (diameter 30 cm, depth 20 cm) were filled with river sand (type 16SS, dimension 0.8/1.2 mm, weight 1.4) and positioned at the center of the growth chamber. A cool white led lamp (V-TAC innovative LED lighting, VT-911-100W, Des Moines, IA, USA) was positioned 50 cm above each seedling, and each plant was grown under an 11:25 h light regime (5:45 a.m. to 5 p.m.). The Photosynthetic Photon Flux Density at 50 cm under the lamp in correspondence of the seedling was 350 μmol_ph_/(m^2^s) (quantum sensor LI-190R, Lincoln, Nebraska, USA). The plants were watered three times a week and fertilized using a half-strength nutrient solution (Murashige and Skoog Basal Salt Micronutrient Solution; see components & organics).

### 4.3. Data Acquisition and Data Processing

For each growth chamber, a pair of RGB-infrared cameras (IP 2.1 Mpx outdoor varifocal IR 1080P) were placed 110 cm above the ground, spaced at 45 cm to record stereo images of the plant (see Figure 1a and Appendix A). The two cameras synchronously acquired a frame every 180 s (frequency 0.0056 Hz). RGB images were acquired during the daylight cycle and infrared images during the night cycle. The anatomical landmarks of interest were the “tendrils” and the “junction” (Figure 1c), developing from the considered leaf. The initial frame was the one corresponding to the appearance of the tendrils and the junction. The final frame was defined as either the frame in which the tendrils start to coil for the “support” condition (number of selected images: 699.62, SD 379.28), or the frame just before the plant fell on the ground for the “no support” condition (number of selected images: 1617.11, SD 1112.82). Images from both left and right cameras were used in order to reconstruct 3D trajectories. An ad hoc software (Ab.Acus s.r.l., Milan, Italy) developed in Matlab was used to identify anatomical points to be investigated by means of markers, and to track their position frame-by-frame on the images acquired by the two cameras to reconstruct the 3D trajectory of each marker. The markers on the anatomical landmarks of interest, namely the tip of the tendrils and the junction, were inserted post-hoc (Figure 1c). The tracking procedures were at first performed automatically throughout the time course of the movement sequence using the Kanade-Lucas-Tomasi (KLT) algorithm on the frames acquired by each camera, after distortion removal. The tracking was manually verified by the experimenter, who checked the position of the markers frame-by-frame. The 3-D trajectory of each tracked marker was computed by triangulating the 2-D trajectories obtained from the two cameras (Figure 1). The 3D coordinates were obtained up to 15 digits after the decimal. The frames corresponding to the time at which the plants grasped the support or touched the ground in the absence of the support were removed from the data set. This was done to prevent classifiers from using these final frames to distinguish between the two conditions. Therefore, the classifiers were only fed with helical organ movements (i.e., circumnutation). Moreover, since each plant has its own starting position, we used the coordinates for the first frame as the origin (0,0,0) for all plants in order to prevent a location bias that could affect learning by the classification models (Figure 3a–c). 

### 4.4. Derived Features

Kinematic variables (hereafter, ‘features’) were analyzed in order to ascertain whether they differed for the “support” and the “no support” conditions. This aspect is fundamental in order to verify the ability of machine learning tools to discriminate across the conditions. To do this, we used the Wilcoxon rank-sum test in R-studio, and the size of the effect calculated as r = z√N where z is the z-score and N is the total number of observations was also considered. In line with previous studies [29,30], we found statistically significant differences between the “support” and the “no support” conditions (Table 3; see also Figure 4 for a graphical example). On the basis of the obtained results, the features considered for model classifications were: (a) junction trajectory; (b) tendril trajectory; (c) junction velocity; (d) tendril velocity; (e) junction acceleration; (f) tendril acceleration; (g) tendrils aperture; (h) overall movement duration; (i) movement duration for each circumnutation; (j) all features (i.e., full kinematics). Please refer to Appendix A for details regarding feature extraction. 

### 4.5. Data Pre-Processing

We adopted a z-score as a data normalization method (standard scaling), by using the formula Z = (x−μ)/σ, where μ stands for the mean value of the feature and σ for the standard deviation of the features. A value equal to the mean of all the features was normalized to 0 and the standard deviation to 1. To avoid biases toward features of the dataset and, at the same time, to prevent the classifiers from learning information from the test dataset, we utilized the transform method to keep the same features from the training data to transform the test data. 

To split the training and test sets, each derived feature was labeled with two different conditions, “support” and “no support”, as a binary labeled dataset. The stratified shuffle split cross-validator was applied to the dataset, which is a merge of StratifiedKFold and SuffleSplit to return stratified randomized folds [17]. The set number of re-shuffling and splitting iterations equals 25, test size as 0.25, default random state.

### 4.6. Models’ Classifications Tasks

The modeling process was carried out with Python. We performed modelling of pea plants’ behavior based on supervised classification frameworks. The purpose of a Machine Learning Classifier is to produce models that, on the basis of a binary-labelled training set, learn to discriminate between different growth circumstances and to provide exact predictions on the basis of an unlabeled test set. Random decision forests (RF), logistic regression (LR), and support vector classifier (SVC) are the classifiers that were applied and compared through cross-validation (see Appendix A) [17]. These approaches are optimized and validated in a wide variety of fields [18,19]. The percentage of test data that were successfully classified for the two conditions is counted under the accuracy of classification. The classification task employed each of the generated kinematic features separately, and the classification accuracy for each feature was evaluated. We also assessed the accuracy of “all features”, where permutation importance was computed following the fitting of the classifiers [31]; we analyzed feature importance for all the derived features. The “overall movement classification” and the “circumnutation classification” are the two broad categories that constitute the model classification task. Each classification task consists of 25 trials, which include 25 iterations of the training and test. The absolute movement duration was typically longer for the plants growing in the presence of a support (Figure 3d) than for the plants growing in the absence of a support. For the “overall movement classification” task, we considered the features extracted from the whole movement for each individual plant (Figure 3**e**). For the “circumnutation classification task”, we partitioned the data into circumnutations, smoothing the data set by generating an approximation function that captured the key patterns, namely the waves of the movement in coordinates (i.e., circumnutation). Then, by cutting between peaks, we split between the waves. The features that were extracted from each circumnutation helped in compensating the dataset for this task. For classifying which condition a single circumnutation corresponded to, the classifiers were fitted to the dataset.

## Figures and Tables

**Figure 1 plants-12-00965-f001:**
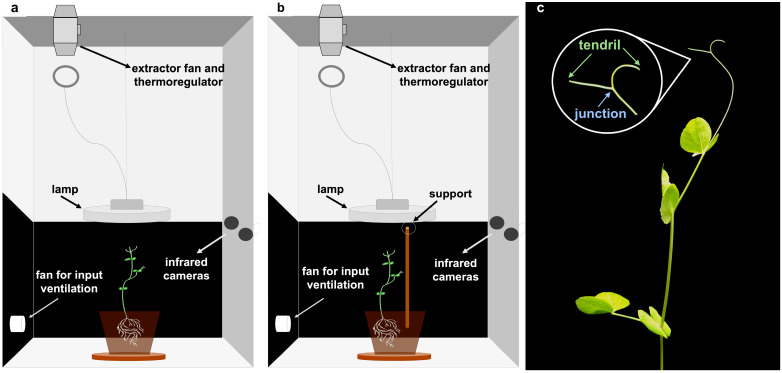
Experimental conditions and anatomical landmarks. Experimental setup, experimental conditions and anatomical landmarks considered. Each chamber is equipped with two infrared cameras on one side, a thermoregulator for controlling the temperature at 26 °C, two fans for input and output ventilation, and a lamp on top of the plant. (**a**) “No support” condition, n = 19. (**b**) “Support” condition, n = 13. (**c**) The anatomical landmarks of interest were the “tendrils” and the “junction” developing from the considered leaf. The “tendrils” refers to the tip of the shoot, and the “junction” refers to the point where the tendrils tie together.

**Figure 2 plants-12-00965-f002:**
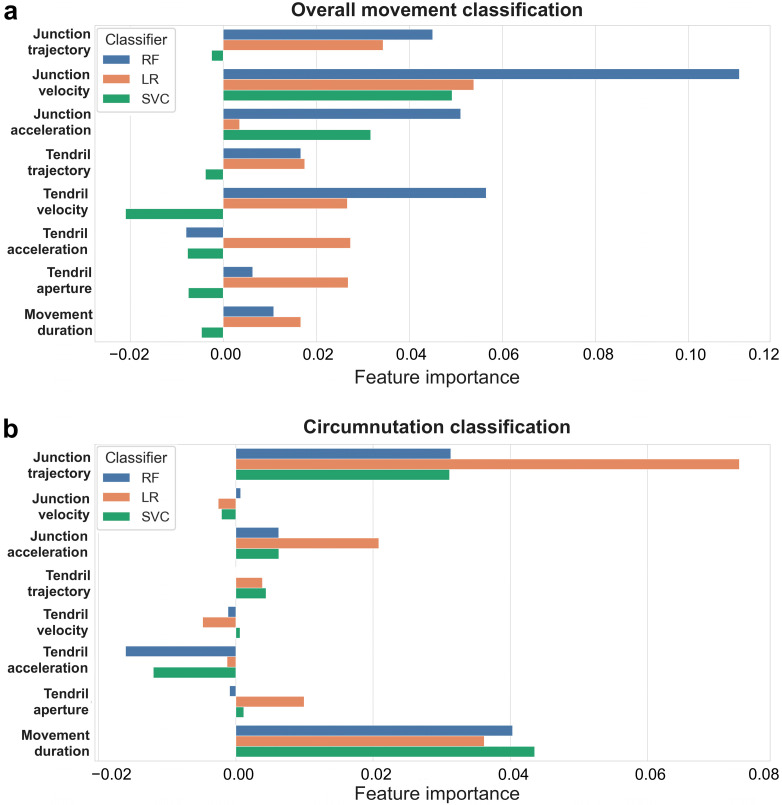
Feature importance in “all features”. Kinematic feature importance of three classifiers random forest (RF, blue), logistic regression (LR, orange), support vector classifier (SVC, green). (**a**) Feature importance in “overall movement classification” task. (**b**) Feature importance in “circumnutation classification” task.

**Figure 3 plants-12-00965-f003:**
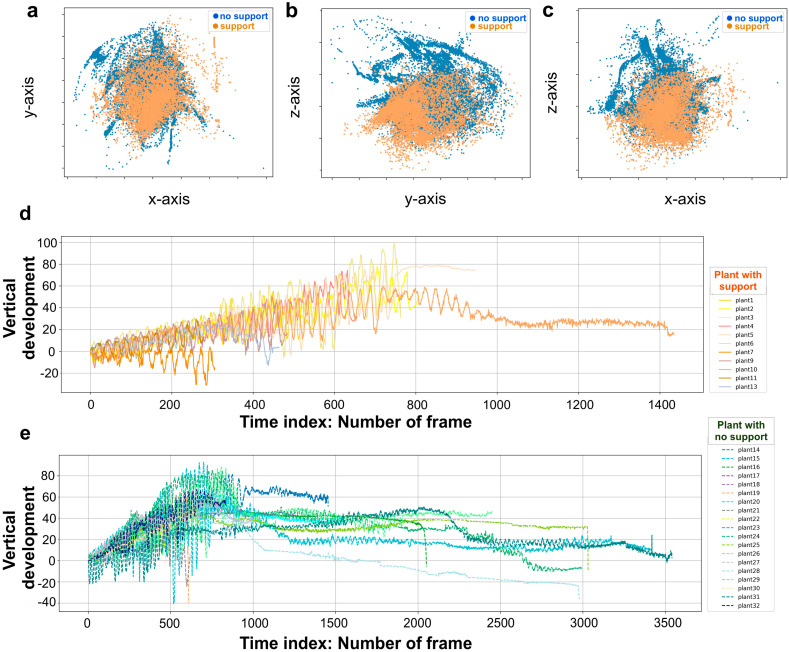
Data acquisition. Coordinates of junction trajectory and plant vertical development in time. (**a**) Junction trajectory for all plants in the x-y dimension for the two experimental conditions. (**b**) Junction trajectory for all plants in the y-z dimension. (**c**) Junction trajectory for all plants in the x-z dimension. (**d**) Junction vertical development in time for the “support” condition. (**e**) Junction vertical development in time for the “no support” conditions. In panels ‘(**d**)’ and ‘(**e**)’, different colors represent different plants. Note that for the “no support” conditions, the length of the time index which is indicated as the number of frames has a longer range than the “support” conditions.

**Figure 4 plants-12-00965-f004:**
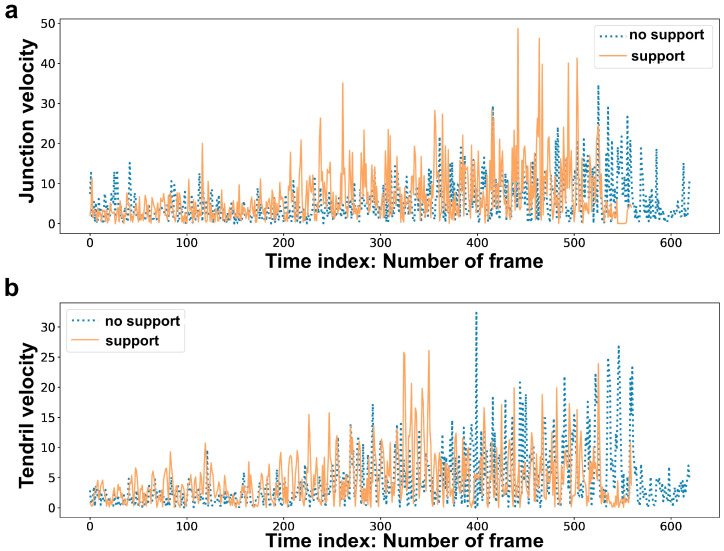
Graphical examples for: (**a**) junction velocity and (**b**) tendril velocity for the “no support” condition and the “support” condition.

**Table 1 plants-12-00965-t001:** Accuracy in “overall movement classification” task. This table shows the mean and standard deviation of the accuracy for each classifier.

	Accuracy % Mean (Standard Deviation)			
	Random Forest	Logistic Regression	SVC	Feature Mean Accuracy
Junction trajectory	71.00 (18.30)	80.50 (13.54)	71.50 (9.89)	74.30 (14.80)
Junction velocity	78.50 (12.24)	78.00 (9.04)	75.50 (12.23)	77.30 (11.19)
Junction acceleration	66.50 (11.81)	72.00 (12.12)	71.00 (11.81)	69.80 (11.99)
Tendril trajectory	67.00 (16.49)	56.50 (14.93)	66.00 (11.13)	63.2 (14.95)
Tendril velocity	75.50 (10.51)	68.00 (15.34)	72.50 (10.21)	72.00 (12.47)
Tendril acceleration	51.00 (11.92)	57.00 (10.87)	63.50 (10.16)	57.20 (12.01)
Tendril aperture	62.50 (15.73)	49.50 (12.23)	60.00 (6.25)	57.30 (13.17)
Movement duration	48.50 (17.43)	65.00 (16.54)	56.50 (10.90)	56.70 (16.48)
All features	76.50 (12.14)	71.00 (13.84)	72.00 (10.38)	73.20 (12.27)
Classifier’s mean accuracy	66.30 (17.36)	66.40 (16.37)	67.60 (11.94)	66.80 (15.39)

Note. A string of accuracy for each classifier and feature is obtained after repeating the classification task 25 times.

**Table 2 plants-12-00965-t002:** Accuracy in “circumnutation movement”. This table shows the mean and standard deviation for accuracy for each classifier.

	Accuracy % Mean (Standard Deviation)			
	Random Forest	Logistic Regression	SVC	Feature Mean Accuracy
Junction trajectory	71.84 (10.71)	74.87 (12.14)	71.54 (14.03)	72.75 (12.29)
Junction velocity	65.09 (11.09)	71.01 (15.23)	70.42 (14.44)	68.84 (13.78)
Junction acceleration	67.12 (9.50)	70.27 (10.44)	69.33 (12.22)	68.91 (10.72)
Tendril trajectory	59.49 (9.10)	68.65 (14.56)	67.38 (12.01)	65.17 (12.61)
Tendril velocity	67.35 (11.39)	70.84 (15.23)	70.37 (14.28)	69.52 (13.63)
Tendril acceleration	62.87 (10.42)	65.62 (12.31)	66.20 (11.23)	64.90 (11.29)
Tendril aperture	64.82 (11.28)	65.60 (11.80)	64.67 (12.79)	65.03 (11.82)
Circumnutation movement duration	63.24 (12.18)	72.98 (12.82)	69.92 (12.58)	68.71 (13.02)
All features	73.74 (12.91)	73.37 (10.35)	72.14 (11.54)	73.08 (11.51)
Classifier’s mean accuracy	66.20 (11.60)	70.29 (12.98)	69.07 (12.96)	68.52 (12.63)

Note. A string of accuracy for each classifier and feature is obtained after repeating the classification task 25 times.

**Table 3 plants-12-00965-t003:** Kinematic data for the “support” and the “no support” conditions. Statistical values obtained when comparing the two conditions are also reported.

	Median				
	No Support	Support	W	*p*	r
Junction velocity (mm/min)	1.7488	3.5035	166	0.007	0.299
Junction acceleration (mm/min)	0.0006	−0.0001	51	0.021	0.257
Tendril velocity (mm/min)	2.5289	4.4670	1242	0.000	0.510
Tendril acceleration (mm/min)	0.0008	−0.0001	361	0.000	0.439
Tendrils aperture (mm)	25.2039	14.7132	245	0.000	0.394
Overall movement duration (min)	3744	1683	59	0.013	0.545
Circumnutation movement duration (min)	201.0857	217.000	143	0.103	0.181

Note. mm = millimeters; mm/min = millimeters by minutes.

## Data Availability

The data is available online: https://doi.org/10.6084/m9.figshare.21215711 (accessed on 17 February 2023). The classifiers used for plant-growing environment identifications and classifications based on pea plants’ movement reported in this paper are available online (https://github.com/qiuranwang/Pea_movement.git, accessed on 17 February 2023).

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
