# Peer review of "Classifying Circumnutation in Pea Plants via Supervised Machine Learning"

_plants, 2023, doi:10.3390/plants12040965_

Round 1
Reviewer 1 Report
Comments for the authors
Manuscript ID: plants-2177623
Title: Unveiling circumnutation in pea plants via supervised machine learning
Wang and co-authors explore the use of machine learning to study circumnutation movements in supported and unsupported plants. They identified different characteristics of the movements and compared them with each other. The paper presents a highly original application of machine learning to plant behavior. I enjoyed reading this manuscript, and I think it will be of interest to a broad audience.
Comments:
I found the abstract very short and a bit cryptic with respect to the content of the manuscript.
Line 32-33. Reference?
Lines 268-269. How many photos were used per plant? Were the photos used in RGB or infrared? Were the photos from the 2 cameras used? This could be included in the supplementary material.
Author Response
Point 1. I found the abstract very short and a bit cryptic with respect to the content of the manuscript.
Response 1. As requested by the Reviewer the abstract has been changed and (hopefully) improved. Please refer to the new version of the manuscript.
Point 2. Line 32-33. Reference?
Response 2. As requested by the Reviewer references have been added in support of the statement reported at line 32-33.
Point 3. Lines 268-269. How many photos were used per plant? Were the photos used in RGB or infrared? Were the photos from the 2 cameras used? This could be included in the supplementary material.
Response 3. All the details requested by the Reviewer have been specified within the new version of the manuscript. Please refer to pp.8.
Reviewer 2 Report
The work may be published in an uploaded form.
Author Response
We thank the reviewer for finding our manuscript publishable in its present form.
Reviewer 3 Report
The research objective was to analyze the circumnutation phenomenon using machine learning techniques. The characteristics of the circumnutation were quantified as some kinetic movements of the stem, and those values were classified and predicted with several machine learning approaches. However, the manuscript must be significantly revised because of the following reasons:
1. The title does not match the manuscript. Since the journal is “Plants,” we could easily think of physiological analysis from “Unveiling”. However, classifying the presence of the support or the circumnutation is not related to the physiological novelty.
2. The section Materials and Methods should be reorganized in the perspective of the English grammar. Several sentences are confusing.
3. The purpose of the research is not that obvious. If the authors wanted to construct a state-of-the-art model to classify the circumnutation, much more kinds of the models such as neural networks should be compared. Otherwise, if the target was the physiological characteristic of the circumnutation, further results such as extracted features of the circumnutation should be represented.
4. Overall, readability of figures should be enhanced. Font sizes are too small.
I recommend narrowing the scope of the study to the classification for circumnutation and clarifying the description.
Author Response
Point 1. The title does not match the manuscript. Since the journal is “Plants,” we could easily think of physiological analysis from “Unveiling”. However, classifying the presence of the support or the circumnutation is not related to the physiological novelty.
Response 1. As requested by the Reviewer, the title has been changed as follows:
“Classifying circumnutation in pea plants via supervised machine learning”
Point 2. The section Materials and Methods should be reorganized in the perspective of the English grammar. Several sentences are confusing.
Response 2. As requested by the Reviewer, an attempt to improve the readability of the ‘Materials and Methods’ section has been made.
Point 3. The purpose of the research is not that obvious. If the authors wanted to construct a state-of-the-art model to classify the circumnutation, much more kinds of the models such as neural networks should be compared. Otherwise, if the target was the physiological characteristic of the circumnutation, further results such as extracted features of the circumnutation should be represented.
Response 3. We thank the Reviewer for these comments. We agree that in order to construct a state-of-the-art model to classify circumnutation a wide spectrum of approaches should be considered. However, regarding the use of neural networks we feel that in small data scenarios as ours using such approach would not be the best course of action. Please note that our data are tabular data and not images. As recently outlined (https://arxiv.org/pdf/2207.08815.pdf) neural networks might not be ideal for tabular data.
As requested, we now provide a graphical representation of how the pattern of junction and tendrils velocity varies depending of the presence/absence of the support.
Point 4. Overall, readability of figures should be enhanced. Font sizes are too small.
Response 4. We apologize for this. The readability of the figures has been (hopefully) improved.
Round 2
Reviewer 3 Report
The manuscript seems to be adequately revised.